# Evaluation of Drought Stress in Cereal through Probabilistic Modelling of Soil Moisture Dynamics

**María del Pilar Jiménez-Donaire [1],\*, Juan Vicente Giráldez [1,2]**  **and Tom Vanwalleghem [1]**

[1]  Department of Agronomy, University of Córdoba, 14071 Córdoba, Spain; ag1gicej@uco.es (J.V.G.); ag2vavat@uco.es (T.V.)

[2]  Institute for Sustainable Agriculture, CSIC, 14071 Córdoba, Spain

\*  Correspondence: g52jidom@uco.es

**Abstract:** The early and accurate detection of drought episodes is crucial for managing agricultural yield losses and planning adequate policy responses. This study aimed to evaluate the potential of two novel indices, static and dynamic plant water stress, for drought detection and yield prediction. The study was conducted in SW Spain (Córdoba province), covering a 13-year period (2001–2014). The calculation of static and dynamic drought indices was derived from previous ecohydrological work but using a probabilistic simulation of soil moisture content, based on a bucket-type soil water balance, and measured climate data. The results show that both indices satisfactorily detected drought periods occurring in 2005, 2006 and 2012. Both their frequency and length correlated well with annual precipitation, declining exponentially and increasing linearly, respectively. Static and dynamic drought stresses were shown to be highly sensitive to soil depth and annual precipitation, with a complex response, as stress can either increase or decrease as a function of soil depth, depending on the annual precipitation. Finally, the results show that both static and dynamic drought stresses outperform traditional indicators such as the Standardized Precipitation Index (SPI)-3 as predictors of crop yield, and the $R^2$ values are around 0.70, compared to 0.40 for the latter. The results from this study highlight the potential of these new indicators for agricultural drought monitoring and management (e.g., as early warning systems, insurance schemes or water management tools).

**Keywords:** drought indicators; drought monitoring; plant water stress; crop yield; Spain

## 1. Introduction

Drought is one of the main natural hazards affecting agricultural crop production and resulting in food insecurity [1,2]. Kim et al. [3] analyzed the global patterns of crop production losses associated with droughts between 1983 and 2009 and concluded that three-fourths of the global harvested agricultural production areas were affected by drought-induced losses. Leng and Hall [4], analyzing global yield losses for different crops under global change, project that yield loss risk will increase in the future. Moreover, their predictions, using an ensemble of models, show that this risk grows non-linearly with an increase in drought severity. Many drought indices focus on the role of water (e.g., SPI, the standardized precipitation index, including only precipitation), although there is some discussion in the scientific community that heat stress might play an equally or even more important role in this yield decline. In a study on historic crop yields in the US, Ortiz-Bobea et al. [5] found an important effect of water stress, although they point to heat stress as the primary climatic driver of future yield changes under climate change. Especially in water-limited environments, however, studies clearly point to drought as the primary driver [6]. However, it is clear that both drought and extreme heat usually occur simultaneously [7]. Lesk et al. [8] estimate a reduction in cereal production across the globe by 9–10% due to the combined effect of droughts and extreme heat. Climate models project

a particularly worrying increase in the frequency and magnitude of extreme events such as droughts for specific areas such as the Mediterranean. This region is considered to be a drought hotspot. Drought is already of great concern today, and climate projections are especially worrying in the Mediterranean [9]. This is combined with the fact that agriculture plays a vital role in its economy, occupying nearly 50% of its total land area. Rain-fed crops are those most likely to come under pressure first by climate change and droughts, although prolonged droughts will also affect irrigated lands and increase the need for more efficient irrigation systems with higher water-use efficiency [10]. Nearly a fifth (21%) of the Mediterranean region is under irrigation, and agricultural water demand represents over 50% of the total water demand in Mediterranean Europe and 81% in Eastern and Southern Mediterranean countries [11].

In order to respond to and control droughts, by managing food resources, planning policy interventions, or assessing agricultural insurance damage [12], it is crucial to assess their impact on agricultural crop yield in a timely and accurate manner. Over the last decades, researchers have developed various drought indices to understand drought intensity and its effects. Meteorological drought indices—for example, the widely used Standardized Precipitation Index (SPI) [13] or the Standardized Precipitation Evapotranspiration Index (SPEI) [14]—have proven very successful but are limited to easily available climatic data. However, these data are often not available at a high spatial resolution due to the sparse distribution of weather stations. Satellite-based drought indices are widely used in agronomic studies; for example, the Normalized Difference Vegetation Index (NDVI) [15] offers a good proxy for vegetation stress. In recent years, several of these indicators were also used simultaneously in combined drought indicators with good results [16,17]. Peña-Gallardo et al. [18] assessed the performance of different meteorological drought indices in Spain for predicting crop yield and found SPI and SPEI to be best correlated with yield. García Leon et al. [19] analyzed a wider range of drought indices and found that satellite-based indices, in particular, the Vegetation or Temperature Condition Indices (VCI/TCI), were able to explain 70% and 40% of the annual crop yield level and crop yield anomaly variability, respectively, for winter wheat and barley.

A better understanding of how drought impacts agricultural production requires comprehending how drought impacts ecohydrological processes. Ecohydrological research has long focused on the interactions and interrelationships between hydrological processes and the structure and function of vegetation, especially its response to drought. However, not much of this research has made its way into the development of drought indicators, which generally focus on either the description of meteorological patterns alone, through meteorological drought indices (e.g., the SPI, rainfall alone, or the SPEI, the Standardized Precipitation Evapotranspiration Index, using rainfall and potential evapotranspiration), or on the observation of the effects of these droughts on plants (e.g., NDVI-based indices). The modulating effect of soil is not generally taken into account in existing drought indices. However, in ecohydrological literature, such a framework does exist and could be very useful for describing the effects of drought on agricultural crop yield. A modelling framework to describe stochastic soil moisture patterns and their effect on vegetation stress was developed in a series of papers by Laio et al. [20], Porporato et al. [1], and Rodriguez-Iturbe et al. [21]. Their research proposed two important indicators, static and dynamic stresses, to assess the effect of drought on plants and the interaction of soils in this process. However, no direct validation of this methodology was performed.

The objective of this paper is therefore to evaluate the use of static and dynamic stresses as a drought indicator. The specific objectives are to (1) calculate static and dynamic stresses for a test area in SW Spain, (2) assess the sensitivity of these two indicators to rainfall and soil conditions, and, finally, (3) validate their performance as predictors of measured crop yield, in comparison to commonly used drought indicators.

## 2. Materials and Methods

### 2.1. Study Area

The study area corresponds to the province of Cordoba, located in the center of Andalusia, SW Spain (Figure 1). This area was selected because yield data were only available at the provincial level (see below). The climate is Mediterranean, with dry, hot summers (Köppen-Geiger climate Csa, [22]). The average annual rainfall for the Cordoba airport station between 1959 and 2018 was 604 mm, with a standard deviation of 243 mm. This high standard deviation illustrates the important interannual variability, with annual rainfall varying between 280 and 1297 mm. The mean annual temperature was 18.0 °C.

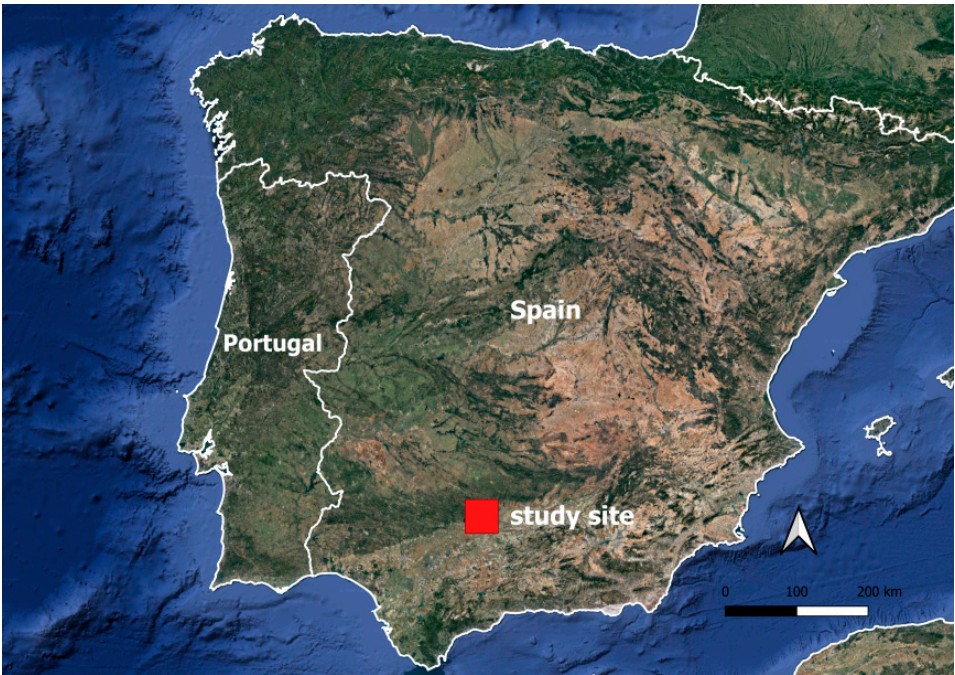

**Figure 1.** Location of study site.

Cereal production in Cordoba province is centered around the Guadalquivir river, and it is part of one of the main cereal-producing areas of Spain [19]. This area is known as the Campiña. The weather station "El Carpio" was selected to be representative for this area (37°54′50″ N, 4°30′14″ W). Soils in the Campiña area are derived from Miocene marls and are typically Vertisols, with a high proportion of expansive clays of ca. 40%. These soils are highly fertile and allow for the typical crop rotation in dryland Mediterranean areas of cereal followed by sunflower during the summer months. Cereal is generally sown during the month of November or early December, depending on the rainfall in that particular year. It is usually harvested during the month of June or early July.

### 2.2. Calculation of Static and Dynamic Stress Indicators

Porporato et al. [1] propose a model framework to quantify vegetation stress related to the soil moisture conditions, based on key concepts of plant physiology. They define a static, $\zeta$, and a dynamic water stress, $\theta$, the latter also taking into account the temporal dimension in the definition of water stress.

The first drought indicator, static stress, was calculated as a function of stomatal closure. Stomatal closure occurs over the entire scale of water stress and starts with the so-called incipient stomatal closure at a soil moisture content of $W^*$. The other end of the scale corresponds to complete

stomatal closure, in which the plant starts wilting, corresponding to a soil moisture content called the permanent wilting point, $W_{pwp}$. Static stress is then defined for the different ranges of soil moisture as

$$
\begin{cases}
\zeta(t) = 0, \text{ for } W(t) > W^* \\
\zeta(t) = 1, \text{ for } W(t) < W_{pwp} \\
\zeta(t) = \left[ \frac{W^* - W(t)}{W^* - W_{pwp}} \right]^q, \text{ for } W_{pwp} \leq W(t) \leq W^*
\end{cases}
\tag{1}
$$

These equations show that static stress is taken as being zero when the soil moisture is above the level of incipient stomatal closure, $W^*$, and that it reaches a maximum value equal to 1 when the soil moisture equals the wilting point. In between these soil moisture values, the vegetation water stress depends on the soil moisture deficit. Plant stress can increase non-linearly with soil moisture deficit, where the coefficient $q$ is a measure of this non-linearity. Porporato et al. [1] suggest that $q$ can vary with plant type and, to a lesser degree, with soil type, although no data exist at present. They suggest a value between 1 and 3, and, in this study, we used a value of 1, implying a linear soil moisture–stress relationship.

The static stress $z(t)$ is calculated at a daily time step. The overall static water stress, $z$, is then calculated by integrating the individual positive values of $z(t)$ over time, excluding periods where $z = 0$. This is because the mean value of water stress should indicate those periods in which the plant is actually under stress, and including nil values without stress in the overall mean would lead to an indicator that is not very informative. In this study, we calculated $z$ over the duration of the growing season; for wheat in the study area, this is between November and June, as will be discussed in detail below. This is because, obviously, only plant water stress in this period has an impact on crop yield. When there is no crop present, the soil moisture deficit cannot contribute to the calculated stress index. The same will be valid for dynamic water stress.

The development of the second indicator, dynamic stress, stems from the realization that the linkage of soil moisture dynamics and plant water stress is a complex problem, due to the stochastic nature of the soil moisture dynamics and the complexity of plant responses [1]. Static stress only takes into account the mean intensity of the plant water deficit but contains no information on its duration and frequency. Therefore a second indicator is proposed, a dynamic stress index that couples the static stress, which represents the integrated effect of the excursion of soil moisture below a critical level $W^*$, with the mean duration and frequency of these stress events, termed, respectively, $T_{W^*}$ and $n_{W^*}$, as follows:

$$
\begin{cases}
\theta = \left( \frac{\zeta \, T_{W*}}{k \, T_{seas}} \right)^{1/ \sqrt{n_{W*}}}, \text{ if } \zeta \, T_{W*} < k \, T_{seas} \\
\theta = 1, \text{ otherwise}
\end{cases}
\tag{2}
$$

where $T_{seas}$ is the duration of the growing season and $k$ is a parameter.

The rationale behind this equation is explained in detail by Porporato et al. [1]. Briefly, the idea behind it is that the same value of $z$ can have a very different effect depending on whether drought occurs as frequent, small episodes or as one, longer episode. For simplicity, it is assumed that a linear relation exists between vegetation stress and the duration of that stress. At present, no data exist to justify a non-linear relation. Therefore, $q$ relates directly to the product of $zT_{W^*}$. However, the actual vegetation stress cannot increase indefinitely with $zT_{W^*}$. The upper value is fixed by the parameter $k$, so permanent plant damage occurs when $zT_{W^*} > kT_{seas}$, and in these cases, the value of $q$ reaches its maximum of 1. The value of $k$ is set to 0.5, following Porporato et al. [1].

## 2.3. Soil Water Balance

To calculate the soil water balance, we followed the same approach presented by Jiménez-Donaire et al. [17]. Soil moisture dynamics are calculated with a simple bucket model, using a volume-balance equation applied over the root zone and taking into account the main processes

of infiltration, evapotranspiration and deep seepage. Therefore, the calculated soil moisture values are representative of the average moisture content over the root-zone depth, $h$.

To evaluate the soil moisture dynamics, the simple water balance model of [23] was used. In this model, the water depth in the soil profile, $W$, evolves with time, $t$, following the contribution of the infiltration of the rain, $f$, and the extraction of the evapotranspiration, $e$, and of the deep percolation or of the surface and subsurface runoff, $g$. The balance was computed at the daily time scale:

$$\frac{dW(t)}{dt} = f - e - g \tag{3}$$

The infiltration depth is estimated from the rain depth, $p$; the wetness or relative soil water content, normalized by the maximum value, $Wmax$, so $\omega = W/Wmax$; and a parameter $m$, with the empirical approximation proposed by Georgakakos [24]:

$$f = p(1 - \omega^m) \tag{4}$$

The deep percolation or runoff loss is estimated by a simple potential function with the saturated hydraulic conductivity, $k_s$, and $\lambda$, the index of pore size distribution of Brooks and Corey [25].

$$g = k_s \omega^{3+2/\lambda} \tag{5}$$

Finally, the daily evapotranspiration rate is estimated as the FAO Penman–Monteith [26] potential rate, $e_0$, modified by the wetness and the crop coefficient, $k_c$:

$$e = \omega k_c e_0 \tag{6}$$

The values for $k_c$ for cereal were set at 0.35 (November to December), 0.75 (January to February), 1.15 (March to May) and 0.45 (June), following recommendations by FAO [27]. The other relevant variables used in the water balance are cited in Table 1.

**Table 1.** Main soil and plant parameters used in the soil water balance and to calculate plant stress.

| Parameter | Value | Source |
|---|---|---|
| $m$ (-) | 0.1 | Mean value of the interval proposed by Brocca et al. [23] |
| $Ks$ (mm day$^{-1}$) | 38.4 | Estimate of soil water properties by Rawls and Brakensiek [28]; representative value for clay soil according to USDA classification |
| $\lambda$ (-) | 0.15 | Derived from graphics of the parameter l of Brooks and Corey [25] as a function of soil texture, organic matter content and increase in soil porosity above the reference [29] |
| $Ws$ (m$^3$/m$^3$) | 0.45 | |
| $W_{fc}$ (m$^3$/m$^3$) | 0.32 | |
| $W_{pwp}$ (m$^3$/m$^3$) | 0.22 | As proposed by Vanderlinden [30] calculated from the soil map of Andalusia |
| $W_r$ (m$^3$/m$^3$) | 0.05 | |
| $W^*$ (m$^3$/m$^3$) | 0.275 | Following Doorenbos en Pruitt [27], taken as 55% of the total available water for cereal |
| $q$ (-) | 1 | Porporato et al. [1] |
| $k$ (-) | 0.5 | Porporato et al. [1] |
| $h$ (m) | 1 | Fan et al. [31] |

## 2.4. Crop Yield Data

Harvest data spanning the years 2003 to 2015 were collected from the Ministry of Agriculture, Fisheries and Food [32], with these statistics being pooled at the provincial level. For this study, focusing on wheat crop yields, we used the data for the Cordoba province, as this is an area representative of one of the main cereal-growing areas in Mediterranean Spain, as mentioned above. The total wheat production area changed over time, from 146,837 ha in 2003 to 84,314 ha in 2015, of which 77–90% is rainfed. The irrigated wheat crop area occupies about 14,000 ha and has remained more constant over this period. It was not taken into account for this study, as it is not likely to be affected equally by drought.

## 3. Results

### 3.1. Soil Moisture Dynamics

Rainfall is very seasonal in the study area, with a clearly defined wet and dry season. It is also highly variable within the study period, with values ranging between 274.8 mm (year 2012) and 853.4 mm (year 2010). This rainfall forcing creates a clearly bimodal probability distribution function of soil moisture, with two marked peaks, shown in Figure 2. The results shown here represents normalized soil moisture values, i.e., $S = (W − W_r)/(W_s − W_r)$, over the entire study period, from 2001–2014. Values of 0 correspond to a soil moisture status equal to residual soil moisture, while values of 1 represent total soil saturation. This figure clearly shows how this bimodal distribution is the resulting sum of a well-marked dry and wet season (respectively, represented in brown, taken from May to October, and blue, taken from November to April). The lower peak, corresponding to the dry season, is close to 0.18, and the other peak, corresponding to the wet season, is around 0.65. The mean overall relative soil moisture is 0.37, with minimum values close to 0 and a maximum value of 0.78. These results provide a good indication that the established water balance model performs well. This is typical for Mediterranean areas, and, although there are no soil moisture measurements available for the study site under cereal, in situ soil moisture observations at a nearby site under olive cultivation showed a very similar bimodal probability density function [33]. These authors also observed a dominating peak corresponding to dry soils for residual water content and another, lower peak at intermediate soil moisture values. To show the variation in soil moisture over the year better, Figure 3 depicts the evolution of normalized soil moisture, S, over the year. In this figure, the mean value of soil moisture is shown in bold, and the gray areas represent the 5 to 95% percentiles, calculated based on daily values of the 2001–2014 period. This figure clearly shows that during the summer dry period, soil moisture drops to a minimum and its variability is about half of that in the wet winter period. This means that all the years analyzed are characterized by an absence of rainfall in this period and a drying out of the soil to values a little above residual soil water content. Around October, the soil moisture starts rising again sharply as the soils are replenished by rainfall. In this period, the variability also rises sharply, as during some wet years, the soil water content is close to its maximum by October, and in other years, the soil moisture remains dry throughout the fall and winter. This can especially be seen in the 5th percentile lower values remaining low. After January, the average soil moisture remains constant until March, after which it drops steadily, although, during wet years, soil moisture can remain high till May, while in dry years, as mentioned before, the soil is never replenished.

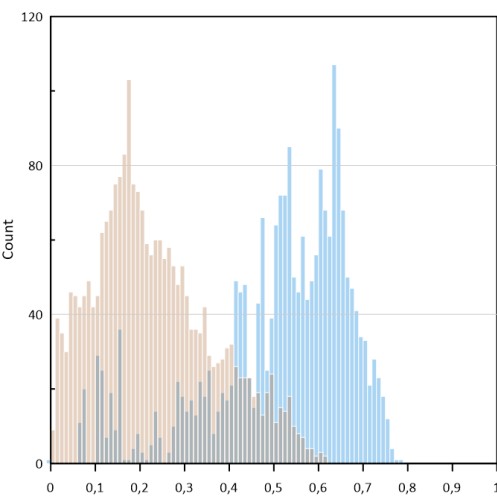

**Figure 2.** Probability distribution of modelled normalized soil moisture, separated into dry and wet seasons (brown and blue, respectively).

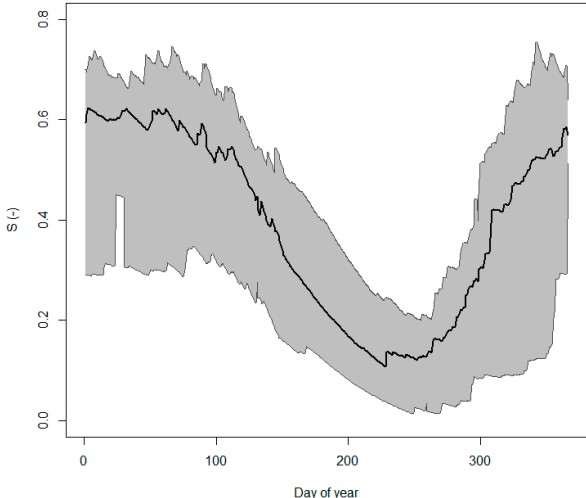

**Figure 3.** Evolution of the seasonal variation of normalized soil moisture, *S*, over the year. Thick line indicates mean values, and gray zone indicates 5 to 95% quantiles.

### 3.2. Static and Dynamic Stress Indicators

Figure 4 shows the evolution of the modelled soil moisture over the study period, in blue, and the resulting static plant water stress, in gray. The extension of each growing season is indicated in green. Static stress generally drops to 0 during the wet winter months and rises to a maximum value of 1 as soon as the soil dries out in spring. Due to the dry Mediterranean summer, it is normal to have a maximum static stress value of 1 outside of the growing season, but these values were not taken into account for the overall yearly calculation. The dry period between 2005 and 2006 is interesting, as these were particularly dry years, and the soil moisture during these years remained low. Therefore, the resulting static stress values remained at a maximum throughout the 2006 growing season. The same happened in 2012. On the other hand, the 2008 growing season was one of the years with the lowest static stress values.

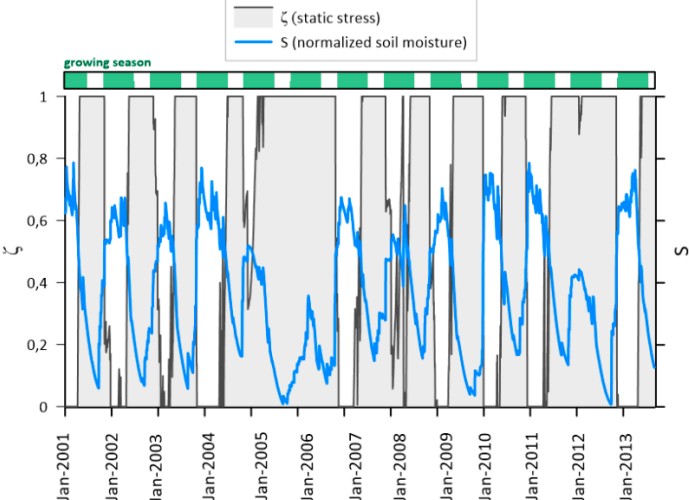

**Figure 4.** Evolution of normalized soil moisture (blue) and static stress (gray) over the study period. Growing season extent is indicated in green.

The dynamic stress is calculated based on the static stress but also taking into account the number and mean duration of the stress events throughout the growing season, as described earlier. This indicator is therefore only calculated once for each growing season. Figure 5 shows how both variables, the number and mean duration of the drought stress events ($n_{W^*}$ and $T_{W^*}$, respectively),

are related to annual precipitation. This figure clearly shows how the dry years are characterized by a single, long stress event. Three years are characterized by a single stress period ($n_{W*} = 1$) that lasts almost the entire growing season (8 months or 243 days). These years correspond to the growing seasons of 2005, 2006 and 2012 (the hydrological years of 2004–2005, 2005–2006 and 2011–2012, respectively), with an annual precipitation of around 300 mm, i.e., less than half the average annual precipitation in this area. Wetter years are characterized by more frequent, but shorter, stress periods. The number of stress periods increased linearly with annual precipitation, while their duration decreased exponentially. In both cases, the fit was significant, although the scatter was high, resulting in a moderate fit with $R^2$ values of 0.40 and 0.50, respectively. The relationship between the number of stress periods and annual precipitation is probably not linear but, rather, characterized by a maximum value and then drops to 0 for higher values of annual precipitation. However, in the study area, this did not occur during the analyzed time period.

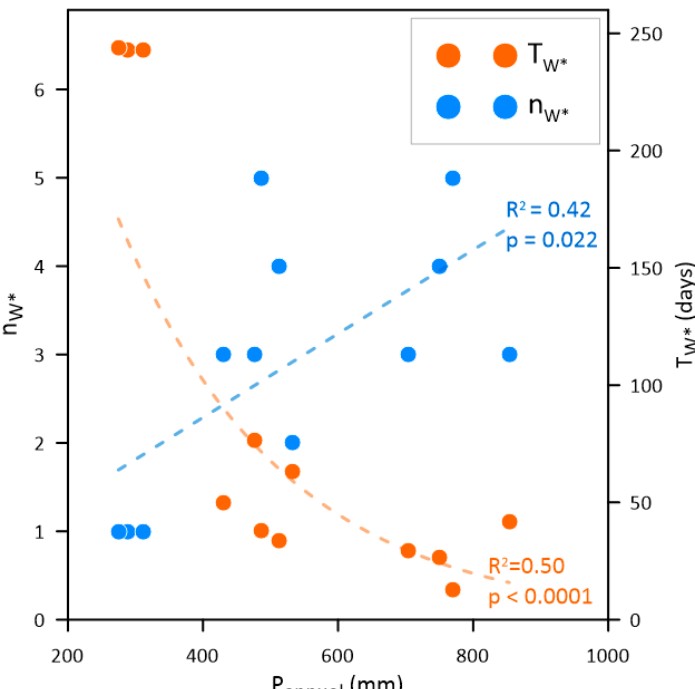

**Figure 5.** Relation of number ($n_{W*}$) and mean duration ($T_{W*}$) of drought stress events with annual precipitation ($P_{anual}$).

Finally, the relation between the static and the dynamic stress is shown in Figure 6 and is fitted by a power relationship. Although this is not the best possible fit in existence, it is used for theoretical considerations, as a power relationship can be deduced from Equation (2). In this equation, a power relation can be derived between static and dynamic stresses if the other variables do not vary too much. Indeed, it can be seen that the values of the product $kT_{seas}$ remain constant for a given crop type, in this case, cereal. The variation in the other two variables, the number and length of drought stress events, $n_{W*}$ and $T_{W*}$, is shown in Figure 5. $T_{W*}$ is generally around 50 days, as most years have multiple short drought periods, except for three years with a single drought that lasts the whole growing season (8 months or 243 days). This power relationship is of interest for characterizing the soil–climate–plant system.

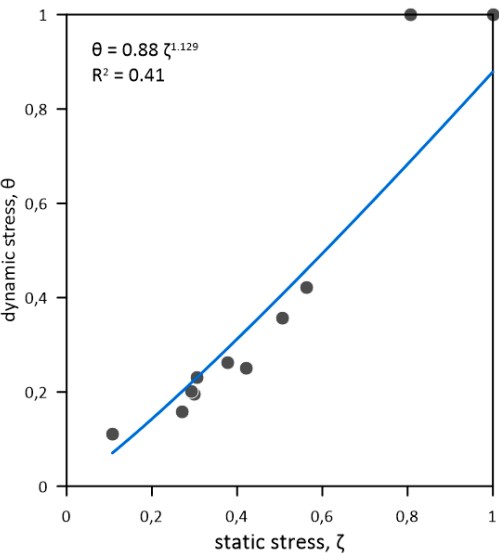

**Figure 6.** Dynamic versus static stress.

### 3.3. Sensitivity of Static and Dynamic Stress Indicators to Soil Depth

As discussed previously, the two stress indicators are closely related to the amount and distribution of the rainfall during the growing season. However, another important variable is the soil type; the soil acts as a reservoir to store water and supply it to the plant when needed. The calculation of soil stress is therefore closely related to the water buffering capacity of the soil, expressed by its total available soil water content. This variable is calculated from the soil depth and from soil water retention behavior, which varies as a function of soil texture and structure. The sensitivity of static and dynamic stresses to the soil water buffering capacity is analyzed here by varying the soil depth. It is assumed here that plant roots can explore the full soil depth, and therefore, soil depth is the limiting variable for root-zone soil moisture storage. This variable is used here to change the soil water-holding capacity, so the same result could be obtained by varying the soil texture or structure, although these variables are not analyzed explicitly. Soil depth can be used as a proxy for both, as, for example, the effect of increasing the pore space would be the same as that of increasing the soil depth. Figure 7 shows the variation of static and dynamic stresses in relation to soil depth. A complex behavior emerges that can be better understood as a function of annual precipitation. Therefore, the different years that fall within the study period were classified into four groups from lower to higher total annual precipitation (brown to blue color). Static and dynamic stresses behave similarly, with small differences that will be analyzed in detail. First of all, for low precipitation values (<431 mm), static and dynamic stresses both increase with soil depth. This increase is gradual for static stress and quite abrupt for dynamic stress. For higher precipitation values (>513 mm), static and dynamic stresses decrease with soil depth. A third group of precipitation values fall in between both behaviors, and first decrease (up to 600 mm soil depth) and then increase.

The reason for the increase with soil depth for lower precipitation values (below 431 mm) is that if both are low, the stress in the system increases for larger soil depths, as the same amount of rainfall results in a lower soil moisture content since the water infiltrates deeper and is averaged out over a larger soil volume. Since the rainfall is so low, there is no benefit from the existence of deep, fertile soils under these conditions. Sites with this soil–rainfall combination would be highly unsuitable for cereal growth. As soon as the rainfall increases, especially for the two classes above 513 mm, it can be seen that deeper soils actually benefit plant growth and reduce plant water stress. The excess soil moisture can be stored under these circumstances, and during dry periods, as long as they are not too pronounced, the soil system can keep up with plant water uptake. The third group (413–513 mm) falls in between both behaviors.

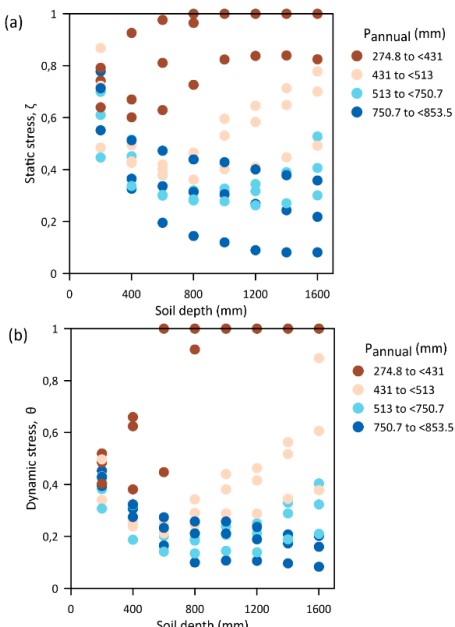

**Figure 7.** Sensitivity of the (**a**) static and (**b**) dynamic stress to soil depth. Color codes show the annual precipitation classified into four levels.

### 3.4. Validation of Static and Dynamic Stresses for Prediction of Crop Yield

It has become clear from previous results that both plant water stress indicators are straightforward to calculate, and the complex response to annual precipitation and soil characteristics has been assessed. The key question that remains is whether these new stress-based drought indicators are of practical use for the prediction of crop yield.

Figure 8 shows the prediction of crop yield as a function of three different drought indicators: the two drought indicators that were evaluated in this study, static and dynamic stresses, and a commonly used drought indicator, SPI-3. Both static and dynamic stresses are shown to be very good indicators, with $R^2$ values of 0.77 and 0.78, respectively. By comparison, SPI-3 performs very poorly. This is surprising given that other studies generally report a reasonable performance of this indicator. It should be noted that one year, marked in red, was considered an outlier. The reason for this is that this point corresponds to the 2006 growing season. During that year, significantly less surface area of rainfed wheat was sown by farmers (−30%), as a response to the bad harvests in the wake of the 2005 drought. This probably resulted in an artificially high average crop yield for that year, compared to other years, as more marginal lands were taken out of production.

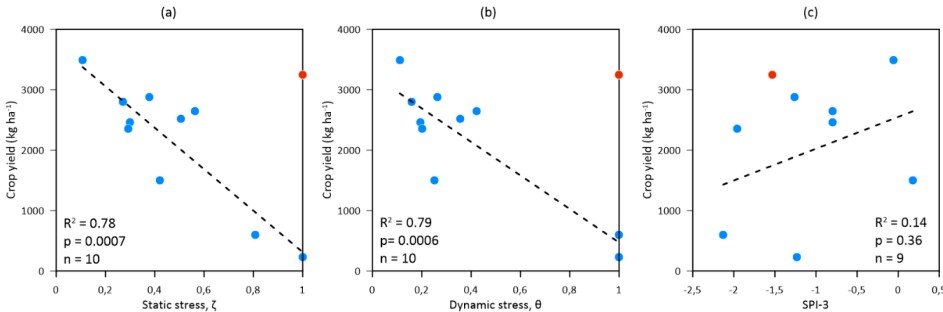

**Figure 8.** Prediction of crop yield as a function of different drought indicators: (**a**) static stress, (**b**) dynamic stress and (**c**) Standardized Precipitation Index (SPI)-3. Note that the red dots indicate outliers that were not taken into account; see text for details.

## 4. Discussion

The results of the sensitivity analysis show a high response of the static and dynamic stresses to soil depth. This shows how important it is to take into account soil properties and plant rooting depth in drought prediction and how both variables can modulate the effect of meteorological conditions on plant water stress. Different soil-moisture-based drought indicators are being developed and tested, although accurate information on soil depth or properties is often missing in these models. Sepulcre-Canto et al. [16] use a Soil Moisture Anomaly index as part of their combined drought indicator. This index is calculated using the LISFLOOD model at a very coarse resolution of 5 km. While useful for continental-scale predictions, the model's simplifications and spatial scale may make it result in a large approximation of the actual soil moisture content and render it less accurate on the farm scale or for agricultural crop yield predictions. A similar multi-indicator approach was tested by Jiménez-Donaire et al. [17], who included a soil moisture deviation as one of the three indicators that made up the drought index and concluded that it corresponded well to agricultural insurance claim data. Narasimhan and Srinivasan [34] developed the Soil Moisture Deficit Index (SMDI), which draws on the hydrological model SWAT, but, again, the spatial resolution is rather coarse (16 km$^2$). Sohrabi et al. [35] developed a specific soil moisture drought index, named SODI, to characterize droughts by calculating the deviation of soil moisture from field capacity. When it was tested in Idaho (USA), the authors concluded that this index outperformed other drought indices such as the standardized precipitation index (SPI), the standardized precipitation evapotranspiration index (SPEI) and the Palmer drought index. However, this is based only on an intercomparison between these drought indices, as they do not use external datasets such as agricultural crop yield data to validate these results. Other promising approaches have relied on the remote sensing of soil moisture rather than modelling in situ soil moisture. For example, Martínez-Fernández et al. [36] developed the Soil Water Deficit Index (SWDI), and Sánchez et al. [37], the Soil Moisture Agricultural Drought Index (SMADI), based on SMOS and MODIS/SMOS products. The remote sensing of soil moisture has the disadvantage that it is only sensitive to superficial moisture [38]; for example, a SMAP radiometer can measure soil moisture up to a 5 cm depth under optimal conditions [39]. However, different studies have shown a good correlation of in situ root-zone soil moisture measurements with remotely sensed superficial soil moisture data [39] or with specifically developed root-zone soil moisture products, such as the 0–100 cm L4_SM that combines the advantages of spaceborne L-band brightness temperature measurements, precipitation observations and land surface modeling with [38,40]. This type of work shows that there is good potential for satellite-based soil moisture drought indices, although, as far as the authors are aware, validation against independent crop yield data, as was performed in this study, has not yet been performed.

Finally, this study also shows the importance of rooting depth for assessing crop sensitivity to drought. This implies that when assessing the agricultural effects of droughts, it is crucial to make specific calculations for different crops. In presently used indicators, this is generally not included, as many drought indicators use reference crop evapotranspiration, and those that take into account soil moisture, a single value for soil properties. This shows that future research should be geared towards combining land use maps with drought indicators to develop specific evaluations for different crop types. In short, our results show that it is of critical importance to correctly estimate root-zone soil moisture in order to calculate drought indices, as both soil depth and rooting depth influence this variable. Our study has performed this through probabilistic modelling because of the long time frame involved, but other approaches using remote sensing products and data assimilation for the estimation of root-zone soil moisture are promising [39,41].

With regard to crop type, it is also important to consider the stage of crop growth. In this study, water stress is currently considered to be equally important throughout the growing season. However, we know that there are certain stages of plant development that are more susceptible than others. Further research could focus on taking this into account, for example, by giving larger weight to drought stress in these periods in the calculation of the overall indicator. However, this escapes the

objectives of this study, whose aim was to test these two simple stress indicators, in the form in which they were designed by Porporato et al. [1]. Further research should aim at perfecting these to obtain even better crop yield predictions.

## 5. Conclusions

This study evaluated two novel indices for drought prediction, static and dynamic plant water stress. These indices are based on early work in ecohydrology by Porporato et al. [1]. Both indices are calculated from simulated soil moisture and take into account the stress that a soil moisture deficit induces on plants. The simulation of soil moisture yields good results, with a bimodal probability distribution that can be clearly divided into two separate populations, one corresponding to the dry season and the other, to the wet season. These results are similar to those of field studies using soil moisture sensors that reported a similar bimodal probability distribution function.

Static and dynamic stresses were shown to detect and correctly quantify the occurrence of dry years in the study period. The number and length of drought periods, two variables taken into account to calculate dynamic stress, were shown to decrease and increase, respectively, with increasing annual precipitation. Both static and dynamic stresses were shown to be highly sensitive to soil depth, and their response behavior, increasing or decreasing, was found to be dependent on total annual precipitation.

Finally, the most important result of this study is that both indicators were found to be good predictors of crop yield. The advantage of these two new indicators, compared to meteorological indices, such as SPI or SPEI, is that the buffering effect of the soil's water holding capacity is taken into account. Therefore, the static and dynamic stresses were found to be superior to the SPI in terms of crop yield prediction, at least in the water-limited conditions of Southern Spain.

In conclusion, both static and dynamic water stress are useful indices for drought detection and quantification. Both indices are easily computed using limited datasets. Where more detailed data are available, these indices can account for the type and depth of soil, in order to calculate spatially distributed drought indices. In addition, they allow one to consider the effect of the length of the growing season and the type of crop by selecting different threshold soil moisture levels for the onset of plant water stress. While this study focused on cereal, further research could focus on evaluating the potential of these indices in other crops or on determining these threshold values for different crops.

**Author Contributions:** Conceptualization, J.V.G. and T.V.; methodology, M.d.P.J.-D.; software, M.d.P.J.-D. and T.V.; formal analysis, M.d.P.J.-D.; investigation, M.d.P.J.-D. and T.V.; writing—original draft preparation, M.d.P.J.-D.; writing—review and editing, M.d.P.J.-D., J.V.G. and T.V.; visualization, M.d.P.J.-D. and T.V.; supervision, J.V.G. and T.V.; project administration, T.V.; funding acquisition, T.V. All authors have read and agreed to the published version of the manuscript.

**Funding:** This study was funded by the research project PID2019-109924RB-I00 financed by the "Programas estatales de generación de conocimiento y fortalecimiento científico y tecnológico del sistema de I+D+i y de I+D+i orientada a los retos de la sociedad".

**Acknowledgments:** The climate data were obtained from the Agroclimatic network of Andalusia ("Red de Información Agroclimática de Andalucía") and were kindly provided by the Instituto de Investigación y Formación Agraria y Pesquera of the Consejería de Agricultura, Ganadería, Pesca y Desarrollo of the Junta de Andalucía.

**Conflicts of Interest:** The authors declare no conflict of interest.

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
