# Peer review of "Evaluation of Drought Stress in Cereal through Probabilistic Modelling of Soil Moisture Dynamics"

_water, doi:10.3390/w12092592_

Round 1

Reviewer 1 Report

The author explored the application of two drought stress indices (static and dynamic stress indexes) over SW Spain to detect droughts. The methodology is well described and the results are well presented and convinced. I only had a minor comment as below:

- In the discussion, the author indicated that it is important to take into account soil properties and plant rooting depth in drought prediction, as both variables can modulate the effect of meteorological conditions on plant water stress. This is true. On the other hand, there is little discussion on the role of root-zone soil moisture in this study. Please authors explicitly discuss this perspective.

- Line 398-400, "Remote sensing of soil moisture has the disadvantage that it only “senses” superficial moisture, ..." 

This is not fully true. There are root zone soil moisture derived from remote sensing surface soil moisture products. See the following literature:

  •  (2020). Quantifying long-term land surface and root zone soil moisture over Tibetan Plateau. Remote sensing, 12(3), 1-20. [509]. https://doi.org/10.3390/rs12030509
  •  others, 2017, Assessment of the SMAP Level-4 Surface and Root-Zone Soil Moisture Product Using In Situ Measurements, Journal of hydrometeorology. 18, 10, p. 2621-2645

- Furthermore, although the English is generally well written, one can find grammar errors. It is highly suggested to have the manuscript English edited.

Author Response

The author explored the application of two drought stress indices (static and dynamic stress indexes) over SW Spain to detect droughts. The methodology is well described and the results are well presented and convinced. I only had a minor comment as below:

We thank reviewer 1 for his positive and constructive feedback.

- In the discussion, the author indicated that it is important to take into account soil properties and plant rooting depth in drought prediction, as both variables can modulate the effect of meteorological conditions on plant water stress. This is true. On the other hand, there is little discussion on the role of root-zone soil moisture in this study. Please authors explicitly discuss this perspective.

Thank you for this comment. In fact, in our study, all soil moisture values actually refer to root-zone soil moisture depth as we make an average soil moisture calculation for the entire root-zone. In this case we selected 1m for the root-zone soil depth, based on data for cereal (see reference by Fan et al., 2016).

Also, when we analyze the sensitivity to soil depth, in fact changing this depth is similar to changing rooting depth. We apologize if this was not clear enough in the previous version and have made changes in the wording of the text regarding the sensitivity analysis to explain this better. In addition, we have extended the materials and methods and discussion on this issue to clarify.

-lines 173-174 “Therefore, calculated soil moisture values are representative for the average moisture content over the root-zone”

-lines 298-299 “It is assumed here that plant roots can explore the full soil depth, and therefore soil depth is the limiting variable for root-zone soil moisture storage.”

-lines 381-385: “In short, our results show that is of critical importance to correctly estimate root-zone soil moisture in order to calculate drought indices, as both soil depth and rooting depth influence on this variable. Our study has done this through probabilistic modelling because of the long time frame involved, but other approaches using remote sensing products and data assimilation for the estimation of root-zone soil moisture are resulting promising [38,39].”

- Line 398-400, "Remote sensing of soil moisture has the disadvantage that it only “senses” superficial moisture, ..." 

This is not fully true. There are root zone soil moisture derived from remote sensing surface soil moisture products. See the following literature:

  •  (2020). Quantifying long-term land surface and root zone soil moisture over Tibetan Plateau. Remote sensing, 12(3), 1-20. [509]. https://doi.org/10.3390/rs12030509
  •  others, 2017, Assessment of the SMAP Level-4 Surface and Root-Zone Soil Moisture Product Using In Situ Measurements, Journal of hydrometeorology. 18, 10, p. 2621-2645

We agree that our statement was overly simplistic. However, reviewer 1 also states himself that these root-zone products are “derived”. We meant exactly the same, but our wording might have been unfortunate. We meant that remote sensing products only sense superficial soil moisture, but indeed secondary products are being generated for root-zone soil moisture. These are however a combination of remote sensing information, modelling and other information sources (such as rainfall data).

The second reference by reviewer 1, Reichle et al (2017) also literally states in his paper “The Soil Moisture Active Passive (SMAP) mission has been providing global observations of L-band (1.4 GHz) passive microwave brightness temperature since 31 March 2015 at about 40-km resolution from a 685-km, near-polar, sun-synchronous orbit (Entekhabi et al. 2010a; Piepmeier et al. 2017). These observations are highly sensitive to surface soil moisture…”

Also Velpuri et al. 2016 state “Optimally, the SMAP radiometer can measure soil moisture up to 5 cm depth.”

So we still consider all this is superficial. It is true however, that superficial soil moisture generally correlates well with deeper, root-zone soil moisture, as is demonstrated by this study by Velpuri et al., 2016 and that modelling and data assimilation has been used to derive secondary products for root-zone soil moisture, such as L4_SM, as stated by Reichle et al., (2017): “data assimilation system that combines the advantages of spaceborne L-band brightness temperature measurements, precipitation observations, and land surface modeling”. But in any case, this is not purely “sensed” by the satellite but a result of complex modelling and using additional information sources in our opinion.

Nevertheless, we agree that we should change the original statement in order to explain this better and have included and discussed the references suggested by reviewer 1.

The new statement is:

“Remote sensing of soil moisture has the disadvantage that it is only sensitive to superficial moisture [38], for example SMAP radiometer can measure soil moisture up to 5 cm depth under optimal conditions [39]. However, different studies have shown a good correlation of in-situ root-zone soil moisture measurements with remotely-sensed superficial soil moisture data [39] or with specifically developed root-zone soil moisture products, such as the 0-100 cm L4_SM that combines the advantages of spaceborne L-band brightness temperature measurements, precipitation observations, and land surface modeling with [38,40],.”

- Furthermore, although the English is generally well written, one can find grammar errors. It is highly suggested to have the manuscript English edited.

We have requested a revision by a native speaker.

Reviewer 2 Report

This is an interesting study, that describes novel indices for predicting cereal yield with variable drought stress. It will be of interest to the scientific community.

The paper is well written, well referenced and the methods are well described. I would recommend this paper for publication.

Author Response

We sincerely thank reviewer 2 for his revision of our paper and his positive recommendation.